# Quantum Mechanical Assessment of Protein–Ligand Hydrogen Bond Strength Patterns: Insights from Semiempirical Tight-Binding and Local Vibrational Mode Theory

**DOI:** 10.3390/ijms24076311

**Published:** 2023-03-27

**Authors:** Ayesh Madushanka, Renaldo T. Moura, Niraj Verma, Elfi Kraka

**Affiliations:** 1Computational and Theoretical Chemistry Group (CATCO), Department of Chemistry, Southern Methodist University, 3215 Daniel Ave, Dallas, TX 75275-0314, USA; 2Department of Chemistry and Physics, Center of Agrarian Sciences, Federal University of Paraiba, Areia 58397-000, Brazil

**Keywords:** hydrogen bonds, hydrogen bond patterns, protein–ligand interaction, semiempirical quantum chemistry, XTB, local mode analysis

## Abstract

Hydrogen bonds (HB)s are the most abundant motifs in biological systems. They play a key role in determining protein–ligand binding affinity and selectivity. We designed two pharmaceutically beneficial HB databases, database A including ca. 12,000 protein–ligand complexes with ca. 22,000 HBs and their geometries, and database B including ca. 400 protein–ligand complexes with ca. 2200 HBs, their geometries, and bond strengths determined via our local vibrational mode analysis. We identified seven major HB patterns, which can be utilized as a de novo QSAR model to predict the binding affinity for a specific protein–ligand complex. Glycine was reported as the most abundant amino acid residue in both donor and acceptor profiles, and N–H⋯O was the most frequent HB type found in database A. HBs were preferred to be in the linear range, and linear HBs were identified as the strongest. HBs with HB angles in the range of 100–110°, typically forming intramolecular five-membered ring structures, showed good hydrophobic properties and membrane permeability. Utilizing database B, we found a generalized Badger’s relationship for more than 2200 protein–ligand HBs. In addition, the strength and occurrence maps between each amino acid residue and ligand functional groups open an attractive possibility for a novel drug-design approach and for determining drug selectivity and affinity, and they can also serve as an important tool for the hit-to-lead process.

## 1. Introduction

Hydrogen bonds (HB)s are the fundamental building blocks of life and are intensively studied non-covalent interactions in biology [1], chemistry [2], and material science [3]. Further, HBs play an indispensable role in structure-based drug designing, with the primitive non-bonding interaction in protein–ligand complexes determining the protein–ligand selectivity and affinity. Studies on the HB nature were conceptually started in 1920, then Linus Pauling reported studies on structural properties of α-helices [4] and β-sheets [5] in proteins to determine the HB geometries in 1951. Until today, intensive studies have been conducted; [1,6,7,8], but the role of HBs in defining structural and biological aspects of drug design is still imprecise. Therefore, a deep understanding of HB geometries and related properties is essential to determine the protein–ligand selectivity and affinity.

Drug potency directly relates to the drug affinity, and for calculating the affinity requires the total strength of protein–ligand interactions. Many HB detection tools, including LigPlot [9], Maestro [10], and BIOVIA Discovery Studio Visualizer [11] are freely and commercially available to investigate protein–ligand interactions. However, the methodology behind these applications are devoted to determining possible protein–ligand HBs from their bond distances and angles, and none of them can take HB strength into account. In this sense, studying variations in HB strength and their relationship with donor/acceptor composition can lead to designing a convenient tool for the pharmaceutical industry.

Known characteristics of HBs can be grouped into four main categories: (1) HBs depend on both acceptor/donor atom pairs and on the atom groups that form the extensive donor and acceptor subunits [12,13]; (2) HBs are composed of attractive electrostatic, induction, and dispersion interactions and repulsive exchange correlation [14]; (3) HBs are weak interactions; and (4) HB length and angle deviate according to their local environments [13,15]. In addition, Bingham and collaborators [16], and Hao [17] pointed out that different HB donors and/or acceptors can significantly influence the HBs strength and structure-activity relationships (SAR) of a ligand and binding pocket could be traced to evaluate HB interactions quantitatively. The strength of a weak chemical interaction such as hydrogen bonding has frequently been investigated via bond dissociation energies (BDE)s [18,19,20], the quantum theory of atoms in molecules (QTAIM) proposed by Bader [21,22,23], the analysis of the natural bond orbitals (NBO) developed by Weinhold [24,25,26]. We added a new method called local vibrational mode analysis (LMA) theory [27,28] originally introduced by Konkoli and Cremer [29].

It is noteworthy that LMA has been utilized for studying systems with a broad range of chemical interactions as described in Refs. [27,28], and turned out to be an excellent tool for the quantitative analysis of chemical bond and weak chemical interactions such as hydrogen bonding. In this context, a python-based program called Efficient Detection of Hydrogen Bonding (EDHB) [30] was developed by our group. EDHB identifies all HBs based on protein geometries, and additionally, protein secondary structure information such as intramolecular ring size and HB networking are also calculated. Further, EDHB provides the functionality of obtaining local vibrational mode force constants and NBO charges through externally correspondent computational packages.

To investigate protein–ligand HBs computationally, an accurate molecular structure must be determined. However, geometry optimization of a protein is computationally challenging because a protein consists of several thousands of atoms [31,32], drastically increasing the consumption of computational time and resources for Hartree-Fock (HF) or density functional theory (DFT) methods. In general, for calculating optimized macromolecular structures at the quantum mechanical level, more approximate approaches such as local fragmentation methods [33], multilevel ONIOM schemes [34], or QM/MM approaches [35,36] and semiempirical methods are commonly applied. Further, semiempirical methods [37] such as DFTB [38], PM3 [39], PM6 [40], and PM7 [41] are also found in the literature for studying protein-based systems.

The primary goals of the present study were (i) to investigate the patterns of HB donor residues, HB acceptor residues, HB types, HB networking and intramolecular HBs; (ii) to study the relationship between HB length and angle; (iii) to explain how protein–ligand HB strengths deviate with HB length, angle and type; (iv) to evaluate HB interactions between ligand functional groups and amino acid residues using average HB strengths and natural occurrences. The overarching goal was to initiate a quantitative structure-activity relationship (QSAR) model to calculate the affinity and predict potential ligand functional groups for a specific binding pocket. In addition to the QSAR model, we aimed to introduce a new technique in the lead optimization process to produce new candidate drug molecules with better pharmacokinetic properties. Finally, to explore if the results obtained in this study can open a new chapter for the prediction of binding affinities and advancement in lead optimization process in structure-based drug designing.

## 2. Results and Discussion

### 2.1. Hydrogen Bond Database (Database A) and Hydrogen Bond Strength Database (Database B)

A protein–ligand HB pattern study requires a large number of HB geometry information from a variety of binding pockets. Thus, we extracted structural information for 11,572 protein–ligand complexes complied in the PDBbind dataset [42,43,44] using EDHB and created protein–ligand HB database A. It is analyzed in terms of their distance, angle, network, intramolecular HB, and relationship with amino acid residues. Database A includes ca. 22,000 HBs. It is imperative to note that many of the protons would be either missing or inaccurate in terms of their geometry due to the various approximations taken for identifying protons in the experimental protein structure determination stage [45,46]. To get an accurate picture of protons, we selected 400 protein–ligand complexes from the PDBBind dataset, added missing protons, and performed geometry optimization and frequency calculation utilizing the XTB in the GFN-xTB level of theory. Then, the HB strength calculations were performed utilizing local mode analysis, and database B was generated, including a total of ca. 2200 HBs. Database B was utilized to study correlations in: (i) HB length and strength, (ii) HB angle and strength, and (iii) interactions of amino acid residues and ligand functional groups. Database A and B consist of important HB information for not only huge numbers but also a large diversity for future drug design projects. PDB IDs for database A and database B can be found in the Appendix A under the topics of PDB IDs for database A and PDB IDs for database B, respectively.

As an example, Table 1 summarizes the HBs identified by EDHB for a protein with PDB ID: **1ofz** [47] shown in Figure 1. A total of 8 HBs were identified, including 4 protein–ligand HBs with Arginine (ARG), glutamic acid (GLU), and Tryptophan (TRP) and four water–ligand HBs. The strongest HB was found to be formed between a water molecule and the ligand with the local mode force constant of 0.329 mDyn/Å. However, the strongest protein–ligand HB was identified to be formed with GLU ($k^{a}$ = 0.311 mDyn/Å) as shown in Table 1. The analysis carried out for database B provides similar information for all 400 protein–ligand complexes.

### 2.2. Amino Acid Residues as Proton Acceptor and Donor

A hydrogen bond donor (HBD) is usually a strong electronegative atom such as N, O, or F that is covalently bound to a hydrogen atom, and a hydrogen bond acceptor (HBA) is an electronegative atom of a neighboring molecule, or ion that contains a lone pair or negative charge. HB formation is determined by both the HBD and HBA characters, and our first task was to investigate the most frequent amino acid residues that can act as HBD and HBA in database A. These occurrences are presented in Figure 2, which depicts the 10 most frequent amino acids in protein–ligand interactions. The reported results point to glycine (GLY) as the most frequent amino acid in both HBD and HBA with frequencies of 24.4% and 41.7%, respectively. Here is highlighted that these are only the first ten most frequent, and all the amino acids appear both as proton donors or acceptors.

In general, GLY is the most frequent amino acid, and serves both as a HBD and HBA. The side chain includes only one hydrogen atom and achiral. It is an essential ingredient in α-helices [48] and collagen triple helices [49]. Further, As a consequence of its small size, it shows compactness, and allows C=O and N–H groups to be more accessible to the interaction with water and ligand molecules.

### 2.3. Hydrogen Bond Types Composition

Figure 3 depicts the occurrence frequencies of different HB types found by EDHB in the entire database A. It is important to emphasize that this composition is accounted for both ligand and amino acids. The major contribution of HBs comes from the N–H⋯O type representing 69.5% of total protein–ligand interactions. Further, O–H⋯O and N–H⋯N show nearly the same percentage (13.9% and 13.2%, respectively). In addition, none of the amino acids are composed of F atoms. However, N–H⋯F occurs only by amino acids as a proton donor (through N–H) and a fluorinated ligand as an acceptor.

An HB is commonly represented as X–H⋯Y, where X is the HBD and Y is the HBA. Both amino acids and ligands can be an HBD. When ligand as an HBD, mainly confined to N–H and O–H ligand functional groups. On the other hand, most amino acids that frequently participate in the protein–ligand interactions such as ARG, HIS, ASP, GLU, and SER, consist of side chains with N–H and O–H functionals (see Mapping HB strength section). This means N–H⋯Y or O–H⋯Y should be predominant. However, HBAs can vary much more in characters such as C=O, O–H, S=O, and P=O making oxygen the major acceptor atom (see Mapping HB strength section). Nevertheless, proteins can form HBs utilizing main chains, and the main chain does not consist of O–H groups due to amino acid polymerization in the protein primary structure (e.g., GLY), which is probably the reason for N–H⋯O predomination.

### 2.4. HB Lengths and Angles Relationship

Goren [50] reported that the deuteron quadrupole constants in O–D⋯O were found to be determined primarily by the O–D bond length and the O–D⋯O angle. Accordingly, we analyzed lengths and angles for ca. 22,000 protein–ligand HBs to investigate important drug-protein patterns. Our HB length and angle graph (Figure 4) shows two clear patterns. HBs found to be in the range of 1.70–2.00 Å corresponds to the angles range of 160–180°, and 2.20–2.40 Å correlates with 100–110°. Further, our results reported that the majority of HBs found in database A have angles in the range of 160–180° showing shorter HB lengths. The results in the present work are in good agreement with those reported by Tan et al. [51], Yunta [52] and IUPAC recommendation 2011 [53]. More recently, Tan et al. [51] reported a study on different types of hydrogen bonds and their preferences of geometry in different environments. Despite defining slightly different angles (R–X⋯X’-H, with $∠RXX^{\prime }$, where X and X’ are heteroatoms), Tan and collaborators reported that protein main chain HBs occur at angle values of ca. 155°, while side chain HBs or HBs with small molecules occur in angle values within 120–130°. In addition, main chain HBs occur at angle values of ca. 105° when exposed to the protein surface. Further, the authors point out that the angle can be explained by interactions between the R–X bond and X’ p-orbital density. In general, saturated X’ p-orbital repeals R-X σ-bond increasing the HB angle. In addition, Yunta [52] showed that when HBs form intramolecular five-member rings, the HB angle could be close to or even less than 110°. Interestingly, this statement matches with our intramolecular results (see Figure 5 in the section of HB network and intramolecular HB) showing 23.0% of five-membered intramolecular HB interactions. Moreover, the third criterion (E3) on the list of the IUPAC recommendations says: “The X–H⋯Y angle is usually linear (180°) and the closer the angle is to 180°, the stronger is the hydrogen bond and the shorter is the H⋯Y distance”. “The X–H⋯Y hydrogen bond angle tends toward 180° and should preferably be above 110° (F4)”. “Historically, the X to Y distance was found to be less than the sum of the van der Waals radii of X and Y. This shortening of the distance was taken as an infallible indicator of hydrogen bonding. However, this empirical observation is true only for strong hydrogen bonds. This criterion is not recommended. In most cases, the distance between H and Y is found to be less than the sum of their van der Waals radii. It should be noted that the experimental distances are vibrational averages and would differ from such distances calculated from potential energy minimization. (F5)” [53].

### 2.5. HB Network and Intramolecular HB

Hydrogen bonds often utilize extensive networks to prearrange binding sites [54,55] and maintain structure and function with a high level of connectivity [56,57]. Sharing the same acceptor or donor atoms of HBs is called HB networking. The networking of HBs can easily allow the required energy for enzyme-catalyzed transformations dictating the HB strengths [58,59]. On the other hand, when the number of potential HBDs increases in a ligand, it can contribute in a negative sense to the protein–ligand binding [60] placing restrictions on its membrane permeability [61]. HB networking can be categorized by the coordination number of the donor, acceptor, or hydrogen(s) from the donor as a string.
(1)$$\begin{array}{c} HBN_{type}=A_{c}-D_{c}-DH_{c} \end{array}$$
where $A_{c}$, $D_{c}$, and $DH_{c}$ denote the number of HBs from the acceptor excluding targeted HB, the number of HB(s) from a donor, and the number of HB(s) from the covalently bonded hydrogen(s) to the donor, respectively. Figure 5a represents HB networking in protein–ligand interactions in database A. Although the majority (59.5%) of protein–ligand interactions do not form HB networks, 0-0-1 (17.4%), 1-0-0 (10.8%), 0-1-0 (9.0%), and 1-1-0 (3.3%) can be identified. The stability of such a hydrogen bond network can be quantified using the robustness, which is qualitatively expressed by Perrin et al. [62] as the ratio of the strength of the network with one HB broken to that of the intact network.
(2)$$\begin{array}{c} r\left(n\right)=1+\frac{k_{B}T\ln\left(n\right)-E_{b}}{E_{b}\;\cdot \;n} \end{array}$$
where $k_{B}$ is the Boltzmann constant, *T* is the absolute temperature, $E_{b}$ is the energy barrier of a single HB, and *n* is the number of HBs. The tightness of a protein–ligand network can vary between 0.0% to 100.0%, and ligands with higher robustness are considered quality lead compounds.

Even though intermolecular HBs mainly contribute for the drug potency, intramolecular HBs also play an important role in the drug development process. When a donor and an acceptor are in proximity on the same molecule, an intramolecular hydrogen bond is formed with a temporary ring system in thermodynamic equilibrium. The open conformation exposes to the solvent, while the closed confirmation covers the polar groups from the solvent. It has been reported that intramolecular HBs are crucial for the favorable orientation of a small molecule in the protein binding site [63,64]. Moreover, conformational preorganization of a ligand for multiple targets confirmed their functions on receptor binding are enhanced with a ligand containing intramolecular HBs [65,66]. Figure 5b shows intramolecular HBs in protein–ligand interaction in database A. The majority of HBs (72.2%) do not form intramolecular HBs. However, 23.0% of HBs form I(5), i.e., five-membered intramolecular rings, and 5.0% even form I(6), i.e., six-membered intramolecular temporary ring systems. The ability of intramolecular ring formation correlates with the acceptor strength in the order C=O > heterocyclic N acceptor > S=O > alkoxy [67] in six-membered rings. Even if intramolecular HBs prefer to be linear, only seven-membered rings have angles in the range of 150–180°. Six-membered rings are in the range of 130–140°, while five-membered rings have long HB lengths and small angles [65].

Organic scaffolds consist of several functional groups such as O–H, N–H, and C=O rendering them soluble and facilitating the formation of specific interactions with their biological targets. When intramolecular HBs are formed, an equilibrium may exist between closed conformations creating a temporary ring system (hydrophobic) and open conformations in which the polar groups are exposed to solvent (hydrophilic). These conformations are not only structurally different but also show distinct physicochemical properties. Further, the closed conformation leads to be more lipophilic [52,68] with higher membrane permeability, whereas the open conformation is water-soluble and has less membrane permeability. In addition to that, it allows conformational restrictions in which the ligand functional groups are favorably aligned with the binding pocket, and the removal of a functional group relevant to the intramolecular ring system can lead to a significant loss of potency [65]. Thus, intramolecular hydrogen bonds play an important role in drug discovery, including techniques such as pharmacophore modeling [69] and biostructure-based drug designing [70]. Therefore, we suggest an important finding for the future development of the pharma industry, HBs found to be in the range of 100–110° and 2.20–2.40 Å show good hydrophobic properties and membrane permeability forming the five-membered intramolecular rings. The dismissal of relevant ligand functional group that generates the ring system can cost significant reduction of potency.

### 2.6. Protein–Ligand Hydrogen Bonds and Strength Relationships

Hydrogen bonding is the main responsible agent for the direction and recognition of substrates by modifying the affinity toward their binding partners. A fundamental understanding of HB strength variations with HB length, angle, and HB local environment can provide the upgradation of contemporary drug designing techniques [15,71]. Therefore, an accurate calculation of the affinity that basically determines by protein–ligand HB strength [72] can greatly contribute to the pharmaceutical industry.

Gibbs free energy difference (Δ*G*) determines the affinity of the drug inside the binding pocket, and fine-tuning of both enthalpy and entropy can achieve high binding affinities according to Equation (Equation 3).
(3)$$\begin{array}{c} \Delta G=\Delta H-T\Delta S \end{array}$$

Here, Δ*H* and Δ*S* are the changes in binding enthalpy and binding entropy, respectively, and *T* is the absolute temperature. According to Equation (Equation 3), strong binding affinity is achieved by either a more negative Δ*H*, a more positive Δ*S*, or the combination of both [73]. Δ*G* also depends on the equilibrium constant, $K_{a}$ (Equation (Equation 4)).
(4)$$\begin{array}{c} K_{a}=e^{-\Delta G/RT} \end{array}$$
where *T* is the absolute temperature and *R* is the gas constant. Accordingly, Gibbs free energy changes lead to a 10-fold higher or lower equilibrium constant due to the exponential relationship [74].
(5)$$\begin{array}{c} \mathrm{HX}+Y<=>X-H\cdots Y \end{array}$$
(6)$$\begin{array}{c} K=\frac{\left[X-H\cdots Y\right]}{\left[HX][Y\right]} \end{array}$$

Equation (Equation 5) shows the reaction of a common HB, X–H⋯Y, and in Equation (Equation 6), *K* is the equilibrium constant for the chemical reaction in Equation (Equation 5). Taft and co-workers [75] developed a strategy to calculate HB strengths using equilibrium constants and proved that the strongest HBA compounds have respectively the largest pK${}_{a}$ values. However, it is not easy to calculate binding energies accurately. As a solution to that, we have introduced a new method of calculating HB strengths accurately using local stretching force constants. The idea of using stretching force constants as a bond strength indicator was initiated by Badger inventing the power relationship between force constants and bond lengths for diatomic molecules in 1934 [76]. However, polyatomic molecules altered from the Badger’s relationship and showed contradictory force constants for spectroscopic analysis because they reflect the coupling between the vibrational modes and depend on the internal coordinates that are used to describe the molecule [77,78]. In 1960, Decius attempted to answer the issue utilizing so-called compliance constants [79] and then, Jones and Swanson from reciprocal compliance constants [80]. In 1998, Konkoli and Cremer [81,82] developed local vibrational modes using a mass-decoupled equivalent of the Wilson equation of vibrational spectroscopy, and later Zou and Cremer [83] proved that a local mode stretching force constant is directly related to the intrinsic strength of a bond recognizing local mode force constants as a unique measure of bond strengths. We calculated local mode force constants for 400 selected protein–ligand complexes and created database B. Using database B, we analyzed HB strength patterns with HB length, angle, and ligand functional groups.

The Badger’s [76] rule failed to explain the power relationship for the polyatomic molecules. As a solution for this, Kraka, Larsson, and Cremer [78] proved the same relationship for polyatomic molecules employing local mode force constants. This is called the generalized Badger’s rule. Figure 6 illustrates the relationship between HB local mode force constants and length exhibiting an apparent power relationship. The results reported here corroborate the generalized Badger’s rule for more than 2200 protein–ligand HBs.

All types of HBs appear along the studied range. Most HBs found to be in the range of 1.70–2.00 Å and local force constant in the range 0.200–0.400 mDyn/Å. However, there are some scattering of data points, especially in the 2.00–2.40 Å region, where local mode force constants are relatively small (local mode force constant 0.000–0.100 mDyn/Å). As can be seen in Figure 6, these points are mainly from N–H⋯O and O–H⋯O, and most of them form HBs with crystal water. It is also observed that O–H⋯F and N–H⋯F are limited in dataset B presenting long HB lengths and small local mode force constants.

The next relationship shows HB strength and angle variation (Figure 7). HB angle results obtained from dataset B corroborate with the trends observed in dataset A. Local mode force constants drastically decrease with HB angles as discussed as a consequence of repulsion between R–X and X’ p-orbital electron densities in R-X⋯X’. HBs found to be in the range of 150–180° correspond with the local mode force constant range in 0.200–0.400 mDyn/Å, while HBs have their angles in the range of 100–150° showed smaller local mode force constants confirming the literature, Wendler et al. [84] suggested that smaller the angle smaller the interaction energy, and IUPAC recommendations [53].

Six HB types were identified from dataset B (Figure 6). Among them, N–H⋯F and O–H⋯F types are uncommon because fluorine is the most electronegative atom, which forms an extremely strong bond with carbon that is challenging to break down. However, recent studies show that F can also be successfully utilized in novel drug developments [85,86]. N–H⋯O, O–H⋯O, N–H⋯N, and O–H⋯N are recurrent in the protein–ligand interactions. Among them, N–H⋯O is the most frequent, and O–H⋯O was observed to be the strongest, while O–H⋯N was the weakest. It is important to point out here we have determined the strongest and weakest HB types depending on the average strength value of each HB type. Although this conclusion contradicts the common belief that nitrogen is a better HBA than oxygen, inside the binding pocket, amino acid residues are not allowed to conformational changes due to the proteins’ secondary and tertiary structure interactions such as HB, ionic, hydrophobic, hydrophilic and etc. In addition, protein–ligand HBs can form networks preventing the conformational prearrangements. Table 2 shows the average strength, length, and angle of each type.

Additional quantum chemical calculations were carried out to evaluate the XTB/GFN2-xTB method with B3LYP-D3(BJ)/aug-cc-pVDZ, described in Table 3 and shown in Figure 8 for (a) NH${}_{3}$ and H${}_{2}$O, (b) water dimer, (c) H${}_{2}$O and NH${}_{3}$ and (d) NH${}_{3}$ dimer. B3LYP-D3(BJ) is widely known for providing reasonable results for HB analysis [87,88], which makes the comparison between these two methods as an initial assessment for validating GFN2-xTB level of theory. Local mode force constants from GFN2-xTB for N–H⋯N and O–H⋯N systems were closely related, and O–H⋯O showed a small deviation with B3LYP-D3(BJ). N–H⋯O deviated considerably from B3LYP-D3(BJ) method showing an artificial shortening in the intermolecular distance and an artificial strengthening of the intermolecular interaction. In addition, B3LYP-D3(BJ)/aug-cc-pVDZ results suggest that N–H⋯O is a weak and unphysical interaction (local mode force constant = 0.013 mDyn/Å and length = 3.3389 Å). This is mainly a result of the basis set superposition error (BSSE) [89] caused by the minimal basis set utilized in GFN2-xTB. Therefore, we removed weak protein–ligand HBs from the study, based on an assessment of both HB length and strength. Apart from that, results obtained at the B3LYP-D3(BJ)/aug-cc-pVDZ level of theory and previous reports [30,90,91] confirm that overall GFN2-xTB is a reliable method for calculating protein-ligand HB strengths.

### 2.7. Mapping HB Strength: Amino Acid and Ligand Functional Group

Knowing the affinity of different amino acids composing the binding pocket with specific ligand functional groups can play an important role in the understanding of potency for a specific target. Further, the same correlation can be utilized to manipulate and obtain the necessary pharmacokinetic properties in the lead optimization process. Figure 9 and Figure 10 depict a correlation between the eight most common ligand functional groups and amino acid residues as natural occurrences and average strengths, respectively, in database B. The protocol presents here provides a HB strength map between different amino acids and ligands regarding their functional groups. Figure 9 illustrates the absolute occurrence of HBs between amino acid residues in protein binding pocket and ligand functional groups of dataset B. Aspartic acid (ASP), glycine (GLY), serine (SER), glutamic acid (GLU), arginine (ARG), asparagine (ASN), and lysine (LYS) were the most frequent amino acid residues, while methionine (MET), phenylalanine (PHE), proline (PRO), cysteine (CYS), alanine (ALA), tryptophan (TRP), and valine (VAL) were the less common counterparts. Further, tyrosine (TYR), leucine (LEU), histidine (HIS), glutamine (GLN), and threonine (THR) appear with a moderate count of HBs.

The most common correlation was identified to be formed between ASP and N–H groups. However, the N–H group can only act as proton donors, and ASP acts as proton acceptors. Furthermore, N–H was identified as interacting frequently with SER, GLY, GLU, and ASN. Figure 11 illustrates the most common N–H correlations highlighting HB counts and average local mode force constants. C=O and ARG, where C=O acts as an acceptor, was the second most common correlation in database B, and also C=O mostly interacted with TYR, SER, LYS, GLU, ASN, and ARG, as illustrated in Figure 12. The O–H group can act as both HBA and HBD. When it is an acceptor, it forms HBs more frequently with GLU and ASP, as can be seen in Figure 13a, while when it is an HBD, it forms HBs with ARG and ASN (Figure 13b). Finally, the P=O functionals most commonly interact with ARG, LYS, THR, and SER (Figure 14).

Due to insufficient information on the occurrence frequency map to explain the correlation between amino acids and ligand functional groups, average HB strengths were calculated for the same ligand functional correlations. The average HB strength map (Figure 10) provides a quantitative image of binding affinities between each amino acid and ligand functional groups. We have categorized average HB strengths or average local mode force constants in 3 convenient ways as category I ($k^{a}$< 0.200 mDyn/Å), II (0.200 mDyn/Å ≤$k^{a}$< 0.300 mDyn/Å), and III ( 0.300 mDyn/Å ≤$k^{a}$). The N–H group forms strong HBs in category II with ASP (0.280 mDyn/Å), GLN (0.274 mDyn/Å), and GLU (0.273 mDyn/Å). The C=O group followed the same trend as N–H with TYR (0.261 mDyn/Å), THR (0.228 mDyn/Å), SER (0.243 mDyn/Å), and ARG (0.211 mDyn/Å). The O–H functional group, when acting as proton acceptors (OH_A in Figure 10) characterized as the strongest HBs with GLU and ASP in category III. On the other hand, the O–H donor (OH_D in Figure 10) was found to be in categories I and II. It is important to emphasize that the O–H tends to form strong protein–ligand HBs when it acts as an HBA. The C–N–C group formed category III HBs with MET (0.295 mDyn/Å) and LYS (0.310 mDyn/Å). However, both the C–O–C and S=O weakly interacted except LYS with S=O in category II (0.253 mDyn/Å). Interestingly, our P=O group was remarkable by forming most of the HBs in categories II and III, including TRP (0.317 mDyn/Å), TYR (0.319 mDyn/Å), SER (0.369 mDyn/Å) and GLY (0.321 mDyn/Å).

The HBs that were identified by EDHB resulted in effective ligand–protein binding in a huge variety of systems (See dataset B in Support Information). In this sense, the protein–ligand interactions that occurred in database B can be considered as a sample of the large population, and the majority of protein–ligand HB interactions form between the amino acids and the eight ligand functional groups that we mentioned. In addition, it is important to emphasize that the drug designing process is very time-consuming and required enormous financial facilities [92]. At the very beginning of the process, one needs to recognize risks before committing significant resources to the project, and one should certify that the target is viable. The ligand-based drug-designing process starts with choosing a viable target and defining its binding pocket. For the known binding pocket, one can determine possible ligand functional groups utilizing our occurrence and strength maps. Further, one can find starting functionals for a pharmacophore model, such as HBAs and HBDs.

In addition to these two, our method can also be used as an important tool in the lead optimization process. The lead optimization process aims at enhancing the most promising compounds to improve effectiveness, diminish toxicity, increase absorption, allow membrane permeability, etc. The higher the quality of lead optimization, the higher the potential for a successful progression into clinical development. However, many of the technologies for lead discovery overlap with lead optimization as researchers attempt to incorporate the best drug characteristics early in the process, which is called Molecular obesity [93]. molecular obesity associated with large MW and high molecular complexity [94]. The occurrence and strengths maps can be utilized to optimize the binding pocket to minimize molecular obesity. We have provided an instance for a hypothetical binding pocket composed of accessible ALA, LYS, and GLU residues. Potential ligand functional groups are described in Figure 15. Here, for ALA: NH, LYS: CNC and GLU: OH_A functional groups can form the strongest HBs generating the highest affinity in the binding pocket.

## 3. Materials and Methods

### 3.1. Dataset and Structure Preparation

The PDBbind dataset [42,43,44] (v.2015) was utilized to carry out the protein–ligand interaction analysis. The dataset includes 11572 proteins with their specific ligands originally from the RCSB databank (https://www.rcsb.org/, accessed on 11 October 2022) experimental structures. This database is maintained and further developed by Prof. Renxiao Wang’s group at the College of Pharmacy, Fudan University in China. In the structure preparation process, binding pockets were separated from the protein within 10 Å sphere distance from the ligand. The *tleap* module of AMBER16 [95] was utilized to add hydrogens and other missing atoms to the experimental structure. Ligand topologies were generated with General Amber Force Field (GAFF) utilizing ANTECHAMBER [96]. The Gaussview [97] and UCSF Chimera [98] software packages were used for visualization.

### 3.2. EDHB: Efficient Detection of Hydrogen Bonding

EDHB by Verma et al. [30] was applied for identifying HBs between protein and ligands. EDHB generates a K-dimension tree (K-D tree) from the coordinates of each hydrogen atom. Donor and acceptor atoms (nitrogen, oxygen, and fluorine) are searched based on geometry. EDHB identifies HB length (1.6 Å to 2.4 Å), HB angle (90–180°), intramolecular HBs, and HB networks. Further, EDHB can calculate local mode force constants and natural bond orbital (NBO) of HBs based on provided natural bond orbital analysis data and second energy derivative information. Most importantly, EDHB outperforms commonly used HBs detecting software in terms of additional features and speed of execution.

### 3.3. XTB Geometry Optimization and Frequency Calculations

The accurate description of macromolecular systems is still one of the challenges in theoretical chemistry. HF and DFT geometry optimizations and frequency calculations are not feasible for large protein systems with a few thousand atoms. On the other hand, molecular dynamic (MD) simulations based on the classical mechanics description are fast even for large biomolecules; however, they cannot describe the breaking/formation of chemical bonds. Therefore, semiempirical quantum mechanical (SQM) methods [37,99,100] have been used to bridge the gap between quantum mechanics (QM) and MD during the last few years. A QM description of the valence electrons and the speed of the calculations (approximately 2 orders of magnitude faster than *ab initio*) qualifies SQM as a reliable and attractive method for the descriptions of macromolecules [101].

In this work, extended semiempirical tight-binding (XTB) methodology [102,103] was applied for geometry optimization and frequency calculations of the binding pocket utilizing the GFN2-xTB [101] (the acronym stands for Geometry, Frequency, Noncovalent, eXtendend Tight-Binding) level of theory. GFN2 includes electrostatic interactions and exchange-correlation effects up to the second order in the multipole expansion and density-dependent D4 dispersion model [101]. Several studies report successful XTB utilization for investigating HB interactions between protein and ligands, including Verma et al. [30] who pointed out that XTB statistically correlates with well-known B3LYP functionals, as well as Ferrero et al. [91] who showed that XTB reasonably agrees with the experimental results.

Local vibrational mode force constants were calculated using the same level of theory that was utilized for the geometry optimization (GFN2-xTB). For comparison, hydrogen bonds in model systems (NH${}_{3}$⋯H${}_{2}$O, NH${}_{3}$⋯NH${}_{3}$, H${}_{2}$O⋯H${}_{2}$O, and H${}_{2}$O⋯NH${}_{3}$) were analyzed utilizing standard DFT methodology, which is reported to reveal a good account for weak noncovalent interactions [104,105]. B3LYP hybrid functional with Grimme’s D3 [106] dispersion corrections and Becke-Johnson damping function [107] was employed with the aug-cc-pVDZ basis set [108].All calculated stationary points were verified as local minima by the eigenvalues of the Hessian matrix. DFT calculations were carried out with the quantum chemical program package Gaussian16 [109].

### 3.4. Local Mode Analysis

To compute the strength of the protein–ligand HBs, we employed the LMA based on the Hessian matrix and its corresponding normal vibrational modes calculated by XTB. LMA was originally developed by Konkoli and Cremer [81,82,110,111] and has recently been described by two comprehensive review articles [27,28]; therefore, in the following, only some of the highlights of LMA are outlined.

Information on the electronic structure and bonding of a molecule is captured in the normal vibrational modes. However, normal vibrational modes are generally delocalized due to kinematic and electronic coupling [77,112]. A certain normal stretching mode between the two atoms of interest can combine with other stretching, bending, or torsional normal modes, hindering the direct relationship between the normal stretching frequency or associated normal mode force constant with bond strength, as well as the comparison between stretching modes in the related molecules. The electronic coupling is generally abolished during a standard frequency calculation following the Wilson GF–formalism [113,114] by transforming from Cartesian coordinates X to normal mode coordinates Q and related normal modes, resulting in a diagonal force constant matrix $K^{Q}$.

Nevertheless, this does not eliminate the kinematic (mass) coupling, which often has been overlooked. Konkoli and Cremer answered this problem by solving mass-decoupled Euler–Lagrange equations [77,81,82,110,111] by setting all atomic masses to zero except those of the molecular fragment (e.g., bond, angle, or dihedral, etc.) carrying out the localized vibration under consideration. The local mode $a_{\mu }$ of a molecular fragment associated with an internal coordinate $q_{\mu }$ is given then by
(7)$$\begin{array}{c} a_{\mu }=\frac{K^{-1}d_{\mu }^{†}}{d_{\mu }K^{-1}d_{\mu }^{†}} \end{array}$$
where $d_{\mu }$ corresponds to a row vector of the normal mode matrix D in internal coordinates $q_{\mu }$ [77].

The local mode force constant $k_{\mu }^{a}$ corresponding to local mode $a_{\mu }$ is obtained by
(8)$$\begin{array}{c} k_{\mu }^{a}=a_{\mu }^{†}Ka_{\mu } \end{array}$$

Other local mode properties such as local mode frequencies, masses, and intensities can be defined accordingly [27,28].

In particular, the local mode force constants k${}^{a}$ have qualified as a quantitative measure of bond strength for both covalent bonds [78,83,115,116,117,118,119,120] and weak chemical interactions, such as halogen bonds, refs. [121,122,123,124,125,126], chalcogen bonds, refs. [127,128,129], pnicogen bonds, refs. [130,131,132] tetrel bonds, ref. [133], and hydrogen bonds [30,134,135,136,137,138,139,140,141,142]. Recently, the quantitative assessment of bond strength in biological systems applying a QM/MM methodology [143,144,145] and the analysis of bonding in actinide and lanthanide compounds have been added to the LMA repertoire [146,147,148]. LMA also successfully revised a number of concepts, such as the characterization of metal-ligand interactions, replacing the Tolman electronic parameter (TEP) with the more accurate metal-ligand electronic parameter (MELP), based on local mode force constants [149,150,151,152,153]. LMA also led to new theoretical insights into an interesting linear relationship between pK${}_{a}$ values and vibrational frequencies [154].

Besides the analysis of bonding, LMA provides a powerful tool for an in-depth analysis of vibrational spectra, as recently applied for the assessment of the usefulness of *vibrational Stark effect* probes [155] or the decoding of characteristic vibrational couplings in nucleobases and Watson–Crick base pairs of DNA [156]. LMA was carried out in this work with the LModeA program package [157].

### 3.5. Pattern Analysis

An HB database (Database A) was generated by executing EDHB for the total number of 11,572 protein pockets in the PDBbind dataset. Further, randomly selected 400 pockets were occupied to generate an HB strength database (Database B) calculating geometry and frequency using XTB in the GFN2-xTB level of theory. Both databases were utilized to investigate protein–ligand HB patterns.

## 4. Conclusions

In this study, we demonstrated that XTB, in combination with LMA, is a reliable technique to investigate protein–ligand HBs and to describe their properties with respect to protein–ligand HB strengths at the quantum chemical level. We generated two pharmaceutically beneficial HB databases, database A including ca. 12,000 protein–ligand complexes with ca. 22,000 HBs and their geometries, and database B including ca. 400 protein–ligand complexes with ca. 2200 HBs and their geometries and bond strengths determined via LMA.

Database A data revealed that glycine is the most frequent amino acid residue for both acceptor (41.7%) and donor (24.4%) profiles, and N–H⋯O (69.5%) is the most common HB type found in the protein–ligand interactions. In addition, we found that protein–ligand HBs form networks such as 0-0-1 (17.4%), 1-0-0 (10.8%), 0-1-0 (9.0%), and 1-1-0 (3.3%) and intramolecular HBs including I(5), i.e., five-membered rings (23.0%) and even I(6), i.e., six-membered rings (5.0%). The majority of HB lengths were found to be in the range of 1.70–2.00 Å and corresponding HB angles in the range of 160–180°. These HBs were identified as the strongest. HBs with bond lengths in the range of 2.20–2.40 Å were found to have HB angles within 100–110°. They usually form intramolecular five-membered ring systems and show good hydrophobic properties and membrane permeability.

Results from database B revealed that the HB local mode force constants and HB lengths of the 2200 HBs in this dataset follow a generalized Badger’s relationship. The correlations found between each amino acid and eight common ligand functional groups can be utilized to predict possible ligand functional groups for a specific binding pocket to generate the highest affinity. Important to note are the novel amino acid and ligand functional group correlations, which resulted in our study, such as the N–H ligand functional group forms strong interactions with ASP, GLN, and GLU as well as C=O with TYR, THR, SER, and ARG, the O–H tends to form strong HBs when it acts as an HB acceptor, and the P=O group was remarkable forming the strongest (category III) interactions with TRP, TYR, SER, and GLY.

Further, we presented an instance analysis for a hypothetical binding pocket composed of accessible ALA, LYS, and GLU residues (see Figure 15), and our results predict that HBs between the N–H ligand functional group and ALA, C–N–C and LYS and O–H_acceptor and GLU provide the highest affinity. In addition, the same correlations can be utilized to determine the starting ligand functional groups for a pharmacophore model, such as HB acceptors and HB donors. Moreover, they can be utilized as an important tool in the lead optimization phase to optimize the binding pocket minimizing molecular obesity. We are glad to mention that the designing process of a QSAR model for the results that were obtained from this study is currently in progress.

## Figures and Tables

**Figure 1 ijms-24-06311-f001:**
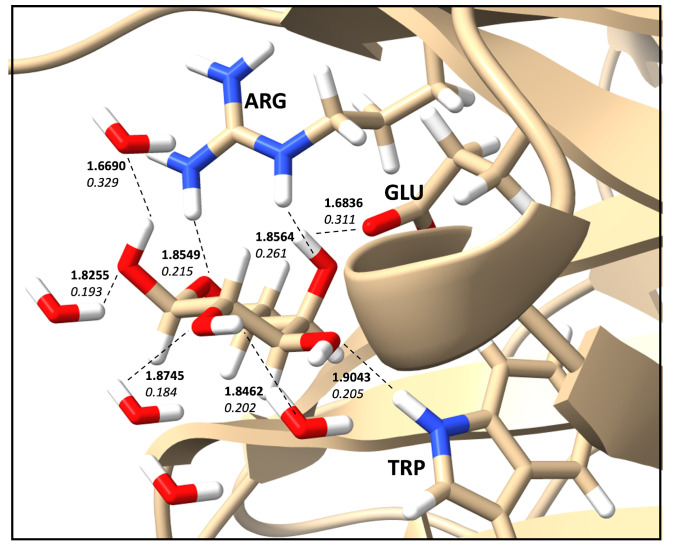
Optimized geometry of PDB ID: **1ofz** with ligand from database B. The binding pocket is indicated by tick sticks, and hydrogen bonds are indicated by dashed lines. The XTB/GFN2-xTB level of theory was used.

**Figure 2 ijms-24-06311-f002:**
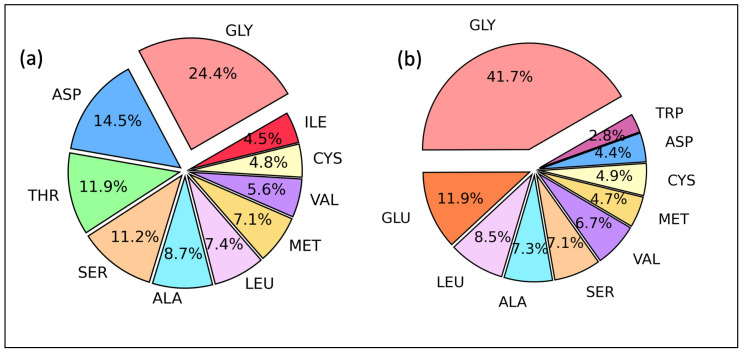
Ten most abundant amino acid residues acting as (**a**) donors and (**b**) acceptors in database A (in percentage). Database A includes ca. 12,000 experimental protein–ligand complexes and ca. 22000 protein–ligand hydrogen bonds with their geometries. Experimental PDB IDs for database A can be found in the Appendix A under the topic of PDB IDs for database A.

**Figure 3 ijms-24-06311-f003:**
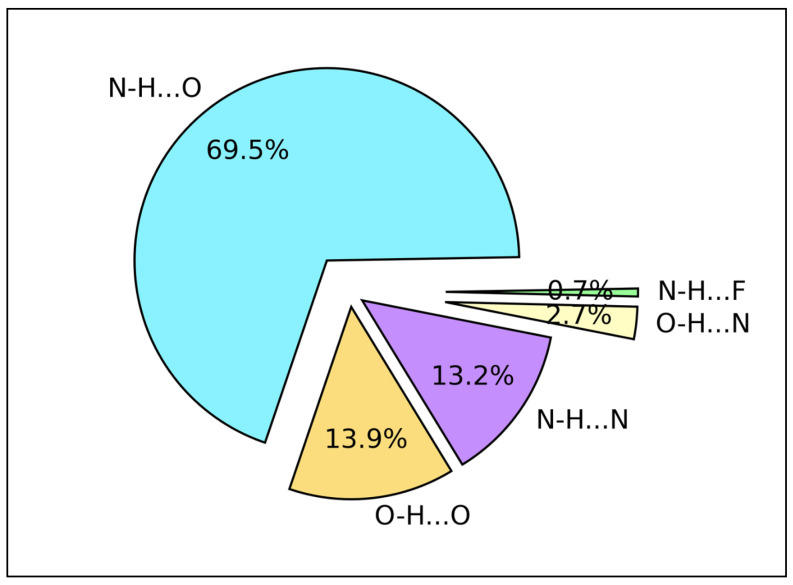
Hydrogen bond type frequencies in database A as X–H⋯Y, where X is the hydrogen bond donor, and Y is the hydrogen bond acceptor (in percentage).

**Figure 4 ijms-24-06311-f004:**
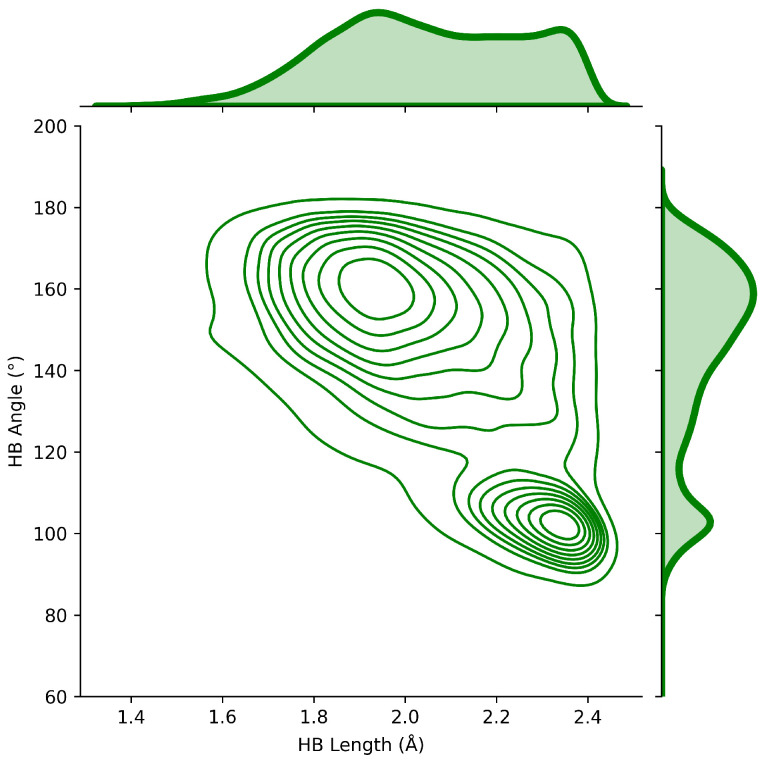
Hydrogen bond length and hydrogen bond angle deviation in database A.

**Figure 5 ijms-24-06311-f005:**
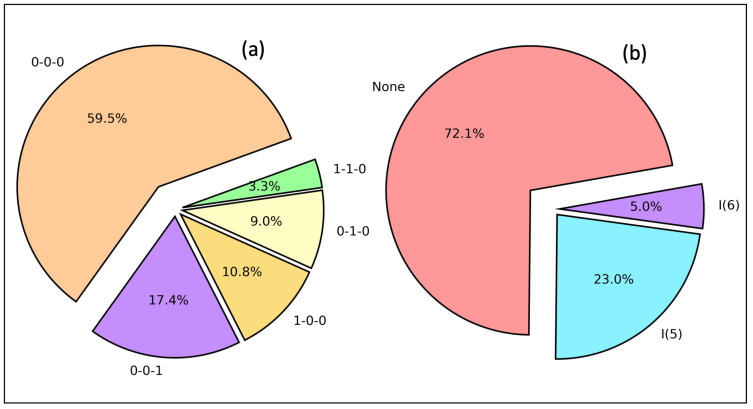
(**a**) Hydrogen bond networking and (**b**) intramolecular hydrogen bonds in database A (For networking, 0-0-0 defined in Equation (Equation 1), and for intramolecular HBs, I(5): five-membered ring and I(6): six-membered ring).

**Figure 6 ijms-24-06311-f006:**
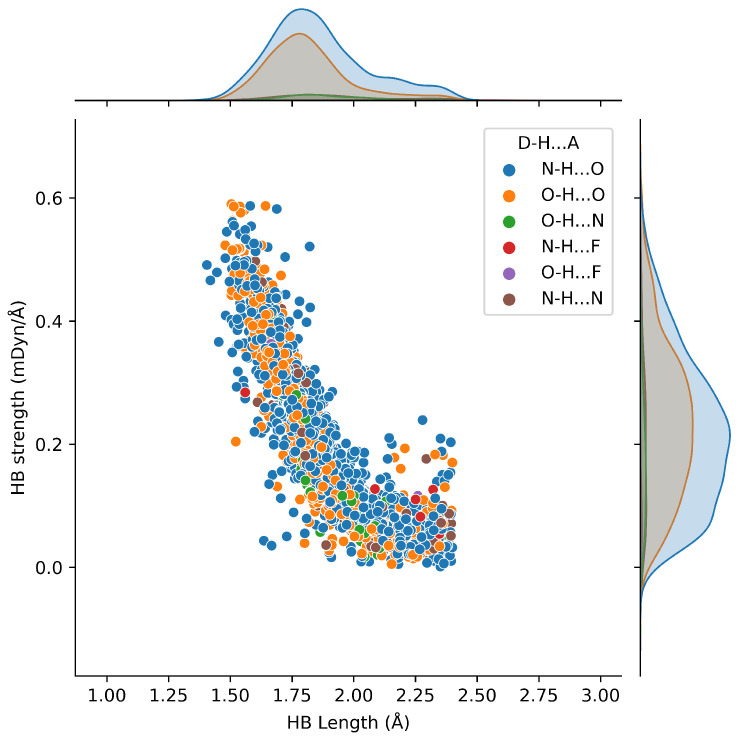
Relationship between hydrogen bond strength and length in database B. Hydrogen bond types describe by the colored points as D-H⋯A, where D is the hydrogen bond donor, and A is the hydrogen bond acceptor. Database B comprised of ca. 400 experimental protein–ligand complexes and ca. 2200 protein–ligand hydrogen bonds with their geometries and strengths. Experimental PDB IDs for database B can be found in the Appendix A under the topic of PDB IDs for database B.

**Figure 7 ijms-24-06311-f007:**
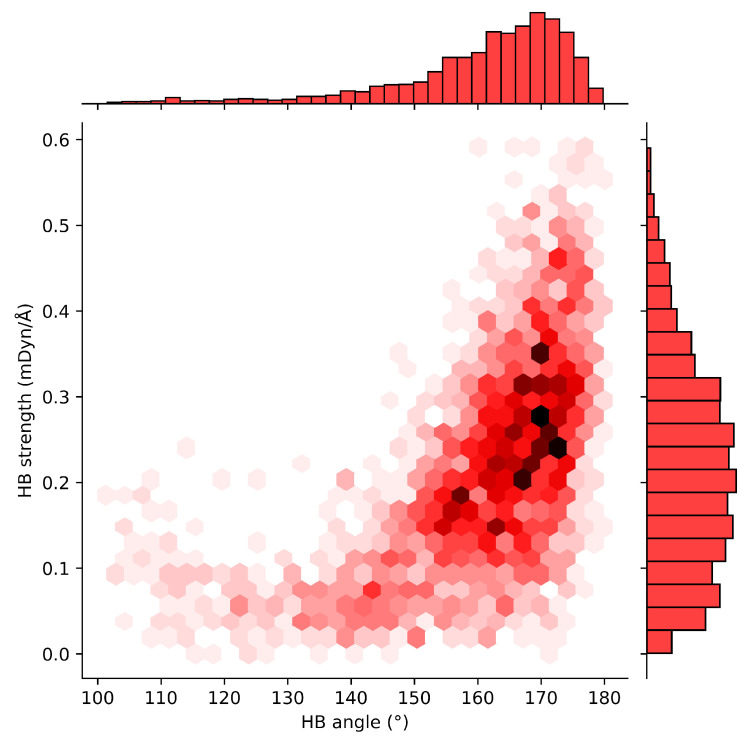
Relationship between hydrogen bond strength and angle in database B.

**Figure 8 ijms-24-06311-f008:**
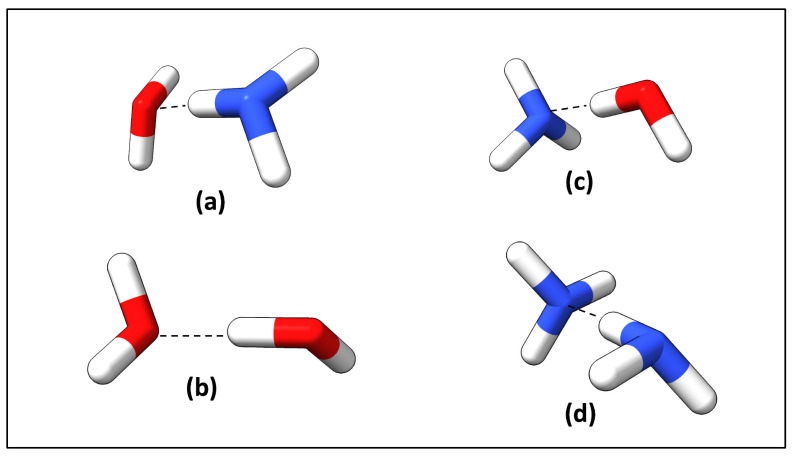
Four HB models prepared for comparison of GFN2-xTB with B3LYP-D3(BJ)/aug-cc-pVDZ. (**a**) N–H⋯O type represents with NH${}_{3}$ and H${}_{2}$O; (**b**) O–H⋯O with water dimer; (**c**) O–H⋯N with H${}_{2}$O and NH${}_{3}$; (**d**) N–H⋯N with NH${}_{3}$ dimer.

**Figure 9 ijms-24-06311-f009:**
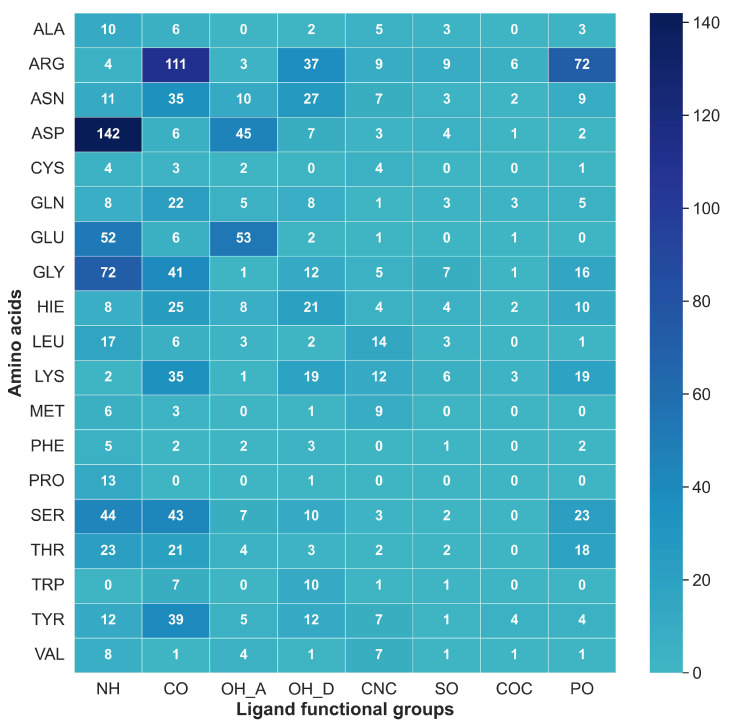
Total number of hydrogen bonds between each amino acid residue and eight most common ligand functional groups in the database B (occurrence map). Ligand functional groups are NH: N–H, CO: C=O, OH_A: O–H acceptor, OH_D: O–H donor, CNC: C–N–C, SO: S=O, COC: C–O–C and PO: P=O.

**Figure 10 ijms-24-06311-f010:**
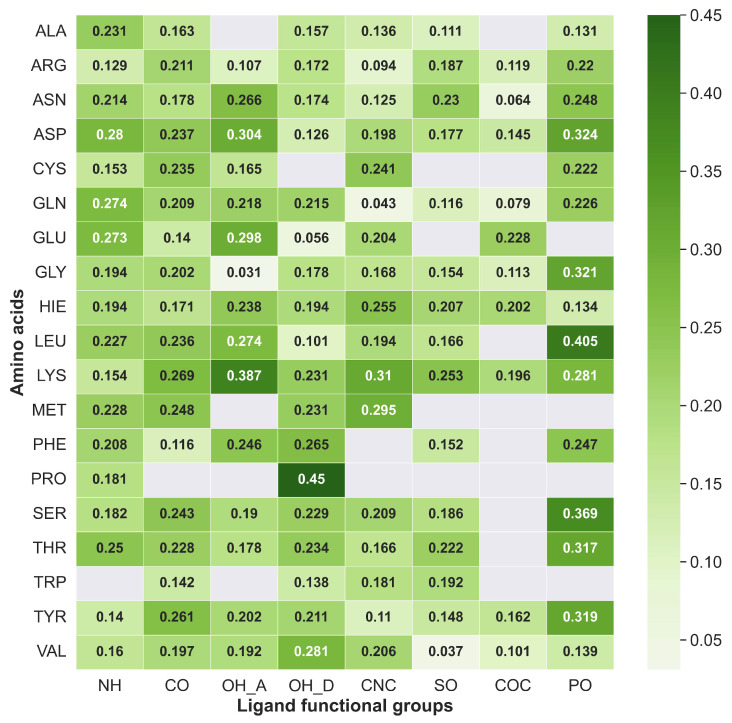
Average strength of the hydrogen bonds between each amino acid residue and 8 most common ligand functional groups (strength map). The white color squares mention a null value representing that these types of hydrogen bonds are not reported in this study. Ligand functional groups are NH: N–H, CO: C=O, OH_A: O–H acceptor, OH_D: O–H donor, CNC: C–N–C, SO: S=O, COC: C–O–C and PO: P=O.

**Figure 11 ijms-24-06311-f011:**
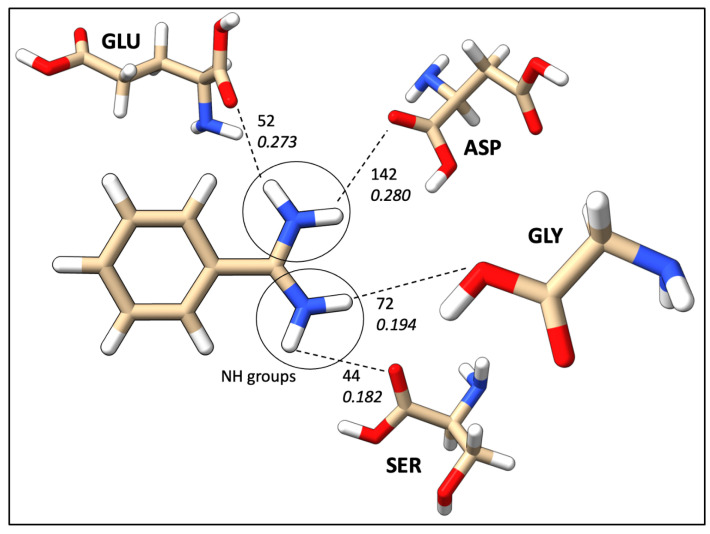
Frequent HBs interactions between the N–H ligand functional group and amino acid residues; the top numerical value mentions the number of HBs, and the bottom value represents the average strength in mDyn/Å from database B.

**Figure 12 ijms-24-06311-f012:**
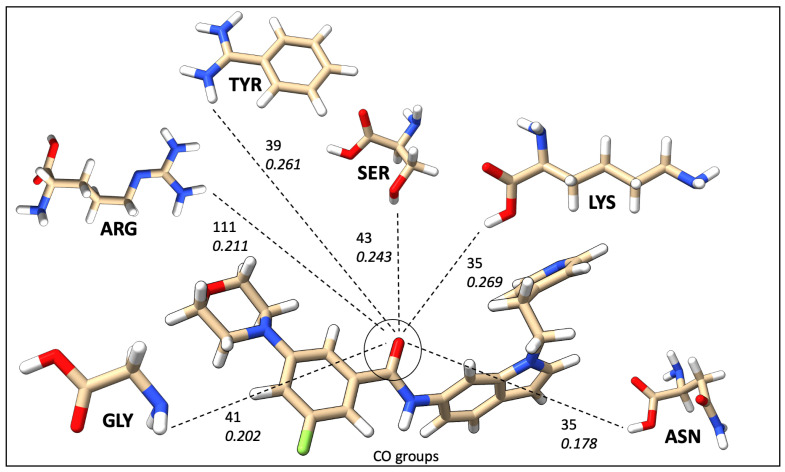
Frequent HBs interactions between the C=O ligand functional group and amino acid residues; the top numerical value mentions the number of HBs, and the bottom value represents the average strength in mDyn/Å from database B.

**Figure 13 ijms-24-06311-f013:**
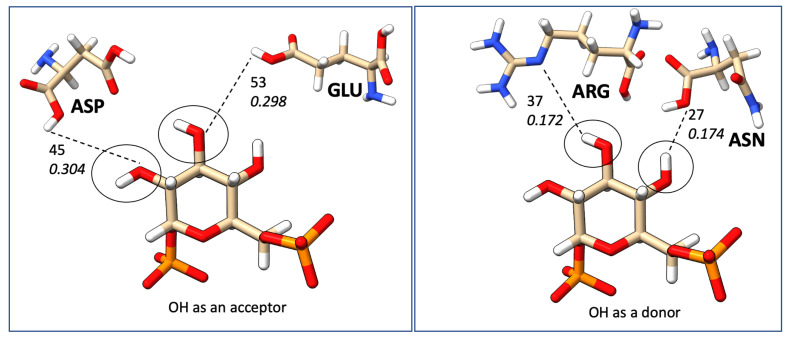
Frequent HBs interactions between the O–H ligand functional group and amino acid residues; the top numerical value mentions the number of HBs and the bottom value represents the average strength in mDyn/Å from database B.

**Figure 14 ijms-24-06311-f014:**
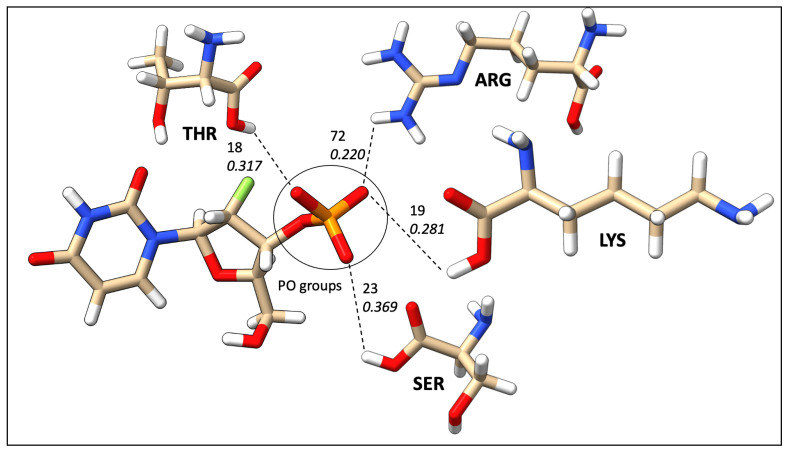
Frequent HBs interactions between the P=O ligand functional group and amino acid residues; the top numerical value mentions the number of HBs, and the bottom value represents the average strength in mDyn/Å from database B.

**Figure 15 ijms-24-06311-f015:**
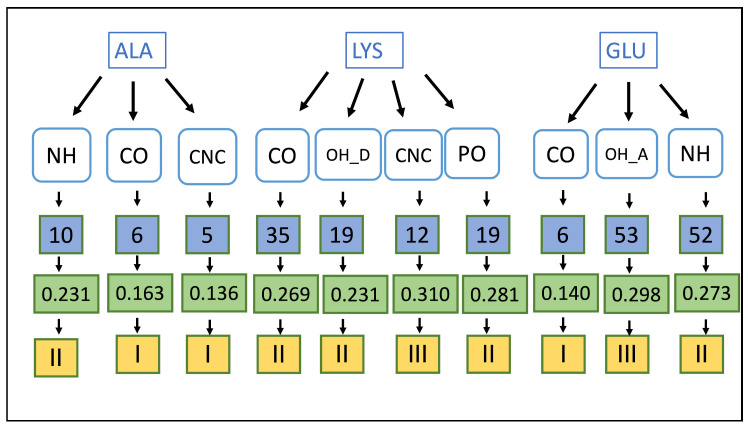
Sample analysis for investigating potential functional groups for a binding pocket with ALA, LYS, and GLU. Blue, Green, and yellow filled shapes represent natural occurrences, average local mode force constants (in mDyn/Å) and HB strength category, respectively. Ligand functional groups are NH: N–H, CO: C=O, OH_A: O–H acceptor, OH_D: O–H donor, CNC: C–N–C, and PO: P=O. HB strength category changes according to: I < II < III.

**Table 1 ijms-24-06311-t001:** Example analysis taken from database B for the PDB ID: **1ofz** with ligand from database B.

Count	Donor ID	Acceptor ID	HB Type	Length (Å)	Angle (°)	Acceptor Force Constants (mDyn/Å)	Donor Force Constants (mDyn/Å)	Intra. HB	HB Network	Donor Residue	Acceptor Residue	Ligand Functional Group
1	O	O	O–H⋯O	1.6836	162.37	0.311	4.352	None	1-0-1	MOL	GLU	OH
2	N	O	N–H⋯O	1.9043	154.20	0.205	6.073	None	0-0-0	TRP	MOL	OH
3	N	O	N–H⋯O	1.8549	156.11	0.215	5.387	None	0-0-0	ARG	MOL	OH
4	N	O	N–H⋯O	1.8564	175.16	0.261	6.052	None	0-0-0	ARG	MOL	CO
5	O	O	O–H⋯O	1.8255	163.02	0.193	6.336	None	1-0-2	WAT	MOL	OH
6	O	O	O–H⋯O	1.8745	175.04	0.184	6.506	None	1-0-2	WAT	MOL	OH
7	O	O	O–H⋯O	1.8462	163.27	0.202	6.231	None	1-0-2	MOL	WAT	OH
8	O	O	O–H⋯O	1.6690	172.92	0.329	4.644	None	1-0-2	MOL	WAT	OH

**Table 2 ijms-24-06311-t002:** Average hydrogen bond length, angle, and local force constant for each HB type in the database B. Database B comprised of ca. 400 experimental protein–ligand complexes and ca. 2200 protein–ligand hydrogen bonds with their geometries and strengths. Experimental PDB IDs for database B can be found in the Appendix A under the topic of PDB IDs for database B.

D-H⋯A (HB Type)	HB Count	< Length > (Å)	< Angle > (°)	< k > (mDyn/Å)
N–H⋯O	1304	1.8598	159	0.219
O–H⋯O	774	1.8150	163	0.234
N–H⋯N	95	1.9310	161	0.195
O–H⋯N	72	1.8993	161	0.171

**Table 3 ijms-24-06311-t003:** Comparison of GFN2-xTB with B3LYP/aug-cc-pVDZ for HB model systems. Local mode force constants (k) are described in mDyn/Å, and length in Å.

Level of Theory	N–H⋯O	Length	O–H⋯O	Length	N–H⋯N	Length	O–H⋯N	Length
XTB/GFN2-xTB	0.069	2.1202	0.192	1.8854	0.093	2.0980	0.206	1.8809
B3LYP-D3(BJ)/aug-cc-pVDZ	0.013	3.3389	0.166	1.9272	0.100	2.2428	0.201	1.9334

## Data Availability

All data supporting the results of this work are presented in tables and figure of the manuscript and in the Appendix A.

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
