# Peer review of "Quantum Mechanical Assessment of Protein–Ligand Hydrogen Bond Strength Patterns: Insights from Semiempirical Tight-Binding and Local Vibrational Mode Theory"

_ijms, 2023, doi:10.3390/ijms24076311_

Round 1

Reviewer 1 Report

In this work, the authors aim to shed light on the patterns of hydrogen bond (HB) donor-acceptor residues and the correlations among bond lengths, angles, and strength in inter/intramolecular HBs by designing two novel HB databases. Because the two databases can provide guidance for protein-ligand selectivity and affinity studies and therefore potentially accelerate the drug design process, they could be invaluable tools for the pharmaceutical industry.

Using a program previously developed by their group, Efficient Detection of Hydrogen Bonding (EDHB), the authors were able to extract structural information from 11,572 protein-ligand complexes in the PDBBind dataset and analyze the binding pockets for HBs (approximately 22,000) and their relevant information. This was used to create their HB database A which includes HB distances, angles, and network information as well as intramolecular HBs and the HB relationships with amino acid residues. The authors also utilized semiempirical quantum mechanical (SQM) methods, namely GFN2-xTB, to carry out geometry optimizations and frequency calculations for 400 randomly selected binding pockets. The strength of each protein-ligand HB, approximately 2,200 total, was calculated using local mode analysis (LMA) and stored in the authors’ HB strength database B. The authors were able to use this data to map the strength between different amino acids and ligands based on their functional groups.

The use of the EDHB program in conjunction with the GFN2-xTB SQM method and LMA appears to be a unique approach to studying HBs and their properties. The two databases the authors have designed are also unique and likely to advance HB-based research, including structure-based drug designing; therefore, the manuscript is recommended for publication pending major text edits and minor technical revisions.

Major Revisions

1) There are a number of spelling and grammar errors throughout the manuscript. A few are mentioned below. These errors were found to detract from the manuscript. As such, it is recommended that the manuscript be carefully proofread before publication.

2) Page 3, lines 131-132: The line “All geometry optimizations and frequency calculations were carried out basis set superposition error (BSSE) corrections” needs to be corrected as it is unclear what the authors intended to say here. Was a counterpoise procedure applied to help correct for basis set superposition error? The CP procedure is well defined for dimers, but additional details need to be provided if any computations were performed on anything larger than a pair of fragments (trimers, tetramers, etc.). 

Minor Revisions

a)    Page 5, line 197 (and throughout the manuscript): The authors are inconsistent with their wording for the Supplementary Materials and also refer to it as the “Supplementary Information.” The authors should consider being consistent with the submitted supplementary document and change all instances of “Supporting Information” to “Supporting Materials.”.

b)    Page 5, Figure 1: It is believed that “The XTB/GFN2-xTB level of theory” was meant to be “The XTB/GFN2-xTB level of theory was used.”

c)     Page 6, Figure 2: The authors should consider editing Figure 2 so the percentages are more easily read.

d)    Page 7, Figure 3: The authors should consider editing Figure 3 so the percentages are more easily read.

e)    Pages 7-9, Figures 3-5: The authors repeat the last two sentences of Figure 2’s caption in the captions for Figures 3 and 4, and it is unnecessary. The authors should consider removing the redundant sentences.

f)      Page 9, line 311: “140°-130°” should be “130°-140°.”

g)    Pages 12, Figure 7: The authors repeat the last two sentences of Figure 6’s caption in the caption for Figure 7, and it is unnecessary. The authors should consider removing the redundant sentences.

h)    Page 13, Table 3: Column 1 refers to aug-cc-pVDZ as a method, but it is a basis set.

i)      Page 13, Table 3 (and throughout the manuscript): The authors should ensure that the number of significant figures used is consistent throughout the manuscript.

j)      Pages 14-15, Figures 9 and 10: The authors repeat the last two sentences of Figure 6’s caption in the captions for Figures 9 and 10, and it is unnecessary. The authors should consider removing the redundant sentences.

k)     Pages 16-18, Figures 12-15: The authors repeat the last two sentences of Figure 6’s caption in the captions for Figures 12-15, and it is unnecessary. The authors should consider removing the redundant sentences.

l)      Pages 16-18, Figures 12-15: The authors should consider increasing the font size of the numbers of HBs and the average strength values.

m)   Pages 19-24, References section: The authors should use a consistent referencing style. Some references have publication years in bold, while others do not (for example, Refs. 62 and 129).

Reviewer 2 Report

In this work, the authors performed a very large scale statistical analysis of H-bonds for a database containing 12000 protein-ligand complexes originally produced by experimental determinations, and for a subset containing about 400 complexes they conducted quantum chemistry calculations to provide deeper insight into H-bond strength. This is a good and informative work, the analysis result is particularly valuable for better understanding H-bond characters in biomolecular systems.

I would like to recommend publication of this work, but revision is needed:

Page 2, Line 41: "HBs are considered electrostatic interactions" this statement is not strict. In fact, dispersion effect plays a comparable role with electrostatic interaction for very weak H-bonds, and polarization effect plays a comparable role with electrostatic interaction for strong H-bonds. See J. Comput. Chem., 40, 2868 (2019) for comprehensive comparison and discussion.

Page 6, Line 218: It is stated that "In General, GLY is the most frequent as both donor and acceptor probably... the side chain can generate only one HB". GLY doesn't have a sidechain, how does GLY form a HB via its sidechain? This point should be clarified.

Page 12, Line 396: "Among them N-H...O is the most frequent and O-H...O was observed to be the strongest, while O-H...N was the weakest" This conclusion is drawn simply via local force constant, however it contradicts with the common belief that N is a better HB aceeptor than O. Binding energies should be given to support this statement, as binding energy is the most rigorous quantity to distinguish strengths of weak interactions.

Page 12, Line 401: I don't agree with the statement "B3LYP is widely known for providing reasonable results for protein-ligand interaction". If dispersion interaction between protein and ligand is non-negligible, then B3LYP performs quite poorly as it cannot represent any dispersion effect. B3LYP-D3(BJ) is a much better choice. I don't know why DFT-D3(BJ) was not employed in this study, it is already well-known that B3LYP-D3(BJ) performs significantly better than B3LYP in studying weak interactions.

In Table 3, the N-H...O length optimized by GFN2-xTB (2.1202) is significantly shorter than that by B3LYP/aug-cc-pVDZ (3.3583), is the data really correct? If they indeed differ with each other so much, I won't believe the GFN2-xTB data in this work makes any sense.

Page 3, Line 120: GNF2 should be GFN2.

Reviewer 3 Report

Madushanka al. reported that Hydrogen bonds (HB)s are the most abundant motifs in biological systems. They play a key role for the determination of protein-ligand binding affinity and selectivity. We designed two pharmaceutically beneficial HB databases, database A including ca. 12000 protein-ligand complexes with ca. 22000 HBs and their geometries and database B including ca. 400 protein-ligand complexes with ca. 2200 HBs, their geometries, and bond strengths determined via our local vibrational mode analysis. We identified seven major HB patterns, which can be utilized as a de novo QSAR model to predict the binding affinity for a specific protein-ligand complex. I recommend this manuscript for publishing in ijms with following some improvements.

Comments:

Ø  Donor and acceptor atoms (nitrogen, oxygen, and fluorine) are 98 searched on the basis of geometry. Why not Cl, in my opinion some cases chlorine also play an important role to establish Hydrogen bond?

Ø  Is this only speed of the calculations make SQM a better method than others?

Ø  How number of HBDs increase On the other hand, when the number of potential HBDs increase in a ligand, it can contribute in a negative sense to the protein-ligand binding  placing restrictions on its membrane permeability. How it should be clear with reason?

Ø  An intramolecular hydrogen bond is formed when a donor and an acceptor are in 298 proximity on the same molecule, forming a ring system in thermodynamic equilibrium, 299 open conformation exposing and closed confirmation covering the polar groups to solvent. It means closed confirmation favours intramolecular HB? Is intramolecular or intermolecular both participate in determining the potency of drug? Need few lines here.

Ø  The Conclusion is should be more appropriate,

Ø  References must be checked again and make sure the consistency in the style of the reference list.

Ø  References are not updated so cite following references

Persistent Prevalence of Supramolecular Architectures of Novel Ultrasonically Synthesized Hydrazones Due to Hydrogen Bonding [X–H⋯O; X=N]: Experimental and Density Functional Theory Analyses. Journal of Physics and Chemistry of Solids, 2021 148, 109679.https://doi.org/10.1016/j.jpcs.2020.109679

A Comprehensive Study of Structural, Non-Covalent Interactions and Electronic Insights into N-Aryl Substituted Thiosemicarbazones via SC-XRD and First-Principles DFT Approach. Journal of Molecular Structure 2021, 1230, 129852. https://doi.org/10.1016/j.molstruc.2020.129852

Non-Covalent Interactions Abetted Supramolecular Arrangements of N-Substituted Benzylidene Acetohydrazide to Direct Its Solid-State Network. Journal of Molecular Structure2021, 1230, 129827. https://doi.org/10.1016/j.molstruc.2020.129827

Reviewer 4 Report

Report on the paper entitled "Quantum mechanical assessment of protein-ligand hydrogen

bond strength patterns: insights from semiempirical tight-binding and local vibrational mode theory", by Ayesh Madushanka, Renaldo T Moura Jr., Niraj Verma and Elfi Kraka, submitted to IJMS.

This paper uses the local vibrational mode analysis developed by some of the authors to investigate the hydrogen bonds in biological,systems. They used a 12000 protein-ligand complexes database, containing 22000 hydrogen bonds, and a second 400 protein-ligand complexes database, containing 2200 hydrogen bonds. Tight binding calculations were performed for the purpose, showing correlations between hydrogen bond length and hydrogen bond angle deviation and hydrogen bond length or angle and hydrogen bond strength. Efficiency of aminoacids interaction was analysed for both acceptor and donor profiles. The most common hydrogen bond type found in the protein-ligand interactions is N-HO.

The paper is well written, with a sufficient references list, and could be published in the present form.

Author Response

We thank this reviewer for his very positive and encouraging report